# Quantity bias in comparison-shopping of multi-item baskets

**Ross Niswanger** [ID]*[◎], **Eric Walden**[◎]

Rawls College of Business, Texas Tech University, Lubbock, Texas, United States of America

◎ These authors contributed equally to this work.

* ross.niswanger@ttu.edu

**Data Availability Statement:** Data for studies in the article can be found in the following: Study 1 - https://doi.org/10.18738/T8/BMKDHI. Study 2 - https://doi.org/10.18738/T8/DHKKZO. Study 3 -https://doi.org/10.18738/T8/G1R7AJ.

## Abstract

Comparison-shopping applications are widespread and have been the subject of considerable research and development. There has also been widespread recognition that people are predictably irrational when making shopping decisions. In this work, we combine these two facts to propose a new type of predicable irrational behavior **that** has important implications for comparison-shopping applications that now utilize crowdsourcing to increase the information provided about sellers in these electronic marketplaces. In a series of three studies we demonstrate that, even after controlling for relative and absolute savings, the number of items in a shopping trip is an important consideration in the decision to make a trip to more than one store. This is true of both actual trips in physical shopping in the real world, and virtual trips to other vendors in online shopping. We term this effect quantity bias.

## Introduction

The growth of the Internet and mobile services has led to increased opportunity for consumers to rely on information technology (IT) enabled shopping [1–5]. Due to the ubiquity of mobile devices, crowdsourcing [6, 7] of information is now possible across a variety of tasks. A particularly interesting new use of technology-enabled shopping is to crowdsource information about prices of goods in physical stores. This has resulted in recommendation agent applications such as Basket®, Favado®, Flipp®, and GasBuddy®, allowing shoppers to do real-time comparison-shopping in physical stores. These applications have significant economic implications for both shoppers and the physical stores at which they shop, as better information should drive down prices [8, 9].

However, research shows that people may violate rational economic principles when evaluating multi-attribute situations [10, 11]. To explain how people violate these principles when analyzing a multi-attribute situation, Thaler [12, p. 183] describes mental accounting as "the set of cognitive operations used by individuals and households to organize, evaluate, and keep track of financial activities" including the formation of topical accounts.

In this work we use the framework of behavioral economics, and in particular the idea of hedonic editing [13–15], which is a type of mental accounting, to examine how multi-item comparison shopping applications are used by consumers. Behavioral economic studies have

**Funding:** The author(s) received no specific funding for this work.

**Competing interests:** The authors have declared that no competing interests exist.

shown that people violate rational economic principles when making shopping decisions [10, 11, 16, 17]. An example of how behavioral economics study this sort of shopping problem is the "jacket and calculator" problem [utilized by 12, 18]:

> *Imagine that you are about to purchase a jacket for ($125)[$15] and a calculator for ($15) [$125]. The calculator salesman informs you that the calculator you wish to buy is on sale for ($10)[$120] at the other branch of the store, located 20 minutes drive away. Would you make the trip to the other store? [12, p, 186 and 18, p. 347].*

Interestingly, a whole cottage industry in comparison shopping applications (e.g. Basket®, Favado®, Flipp®, and Walmart Saving Catcher®) has grown up to help consumers engage in exactly the type of economic decisions that behavioral economists study in experiments like the jacket and calculator problem. Specifically, how much would a person have to save to travel to the other store? Shopping comparison apps have the potential to disrupt physical shopping by providing customers information on how much they could save and hence enabling this decision in near-real-time.

However, behavioral economics literature has identified some biases that occur in making this traveling-to-a-store problem. People do care about the amount of money they save by traveling to a store, but they also care about the relative amount of money they save, so they are more likely to travel to the second store to save $5 on a $15 calculator than to save the same $5 on a $125 jacket. This is an example of predictable irrationality [19]. Thus, while people may not follow strict economic rules, their methods of economic decision making are consistent and predictable, so they can be modeled. Research needs to discover these economic decision-making rules and empirically validate them. That is our goal in this work.

Taking a trip to the grocery store is something that consumers can do as a fill-in trip or as a major trip [20–22]. A fill-in trip is a trip for one or few items. A major trip is a trip to purchase a diverse set of items for that one trip vs. a fill-in trip that is to satisfy an urgent need and usually has a smaller set to fill. A fill-in trip that only requires a single item is the situation that has been studied so well in behavioral economics [23–26]. However, our interest is in major trips that are supported by applications, which allow users to construct a basket that is distributed across multiple stores.

These shopping basket comparison applications have one major difference from the questions behavioral economists typically examine. Real world shopping baskets typically contain multiple items and hence shopping basket comparison applications help the user optimize a multi-item basket rather than just a single item. One under-investigated type of economic decision making may concern how multi-item savings is calculated, which is an important consideration because the actual decision question people typically face is traveling to a second store to save money on some subset of the items they wish to purchase vs. purchasing all the items at one store. While traveling to more than two stores is an option when filling a multi-item basket, we have bounded this initial question of determining if a bias exists by examining the relationship between deciding to go to one or two stores. In this paper we examine the question of how behavioral economics applies to a basket of items and we conduct a series of studies to demonstrate a novel decision rule that applies to these sorts of multi-item shopping situations, which we term quantity bias.

In these studies, we demonstrate the existence of a new economic decision bias, "quantity bias", in which people are less willing to go to a second store when fewer items are to be acquired. This is similar to relative value bias [23–26] in which people are less willing to go to a second store when the relative savings is low. In this paper, we address this issue of quantity as

a type of account that predisposes consumers to making non-normative decisions under certain conditions.

## Theoretical framework

### Absolute savings

Assume a situation where a user has a multi-item basket of products, which they want to purchase. For simplicity we focus on commodity items that are identical at different stores. Additionally, while other types of biases are probably at work when considering these price-comparison applications, the number of items at each store is a basic information piece. We designed away other biases that might be in play that we could, as we wanted to isolate the effect of the number of items compared to savings.

Using a representative application (see Fig 1), consumers specify the list of items they desire and then the application calculates the lowest cost for the entire list based upon the number of stores that the shopper is willing to travel to. Current applications have options for a single store, two, or three or more stores, but for simplicity we limit our enquiry to two stores. Thus, the user must decide between two alternatives. The user can simply go to the store that has the overall lowest cost, or the user can make a second trip and get all of the items at store 1 that are less expensive at store 1 and all of the items at store 2 that are less expensive at store 2. The implementation of this in a representative app is shown in Fig 1. The left panel shows that the cost at the least expensive store is $28.01. The right panel shows that the total cost is reduced to $26.33 if two stores are utilized for those same items (the bread is less expensive at Red Circle than Blue Diamond).

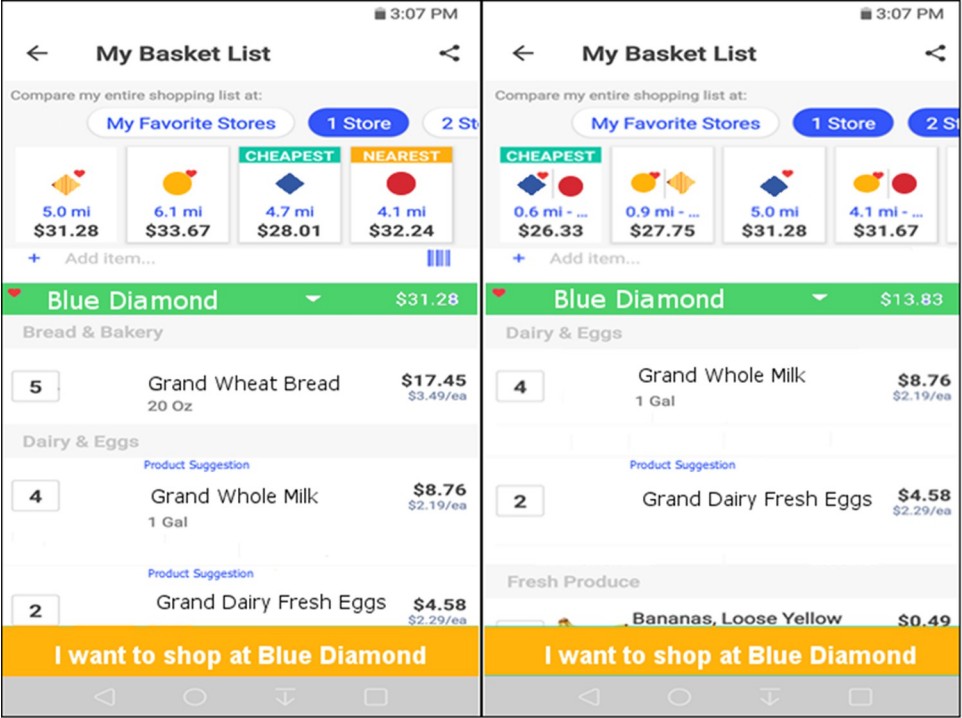

**Fig 1.** Screenshots from a representative application illustrating the effect on total cost by going to one store (left hand side) or two stores (right hand side) to complete the purchase of a basket of items. Total cost is lowered from $28.01 to $26.33 if the user chooses the cheapest two-store option.

In principle, a rational decision maker will only consider the cost/benefit of making a trip to a second store. Thus, if it takes 10 extra minutes to travel to the store, the rational question is "how much is 10 minutes of my time worth?" However, people have been shown to be irrational when making the decision to drive to a second store. Moreover, they have been found to be *predictably irrational* (Thaler 1999, Ariely 2010). That is to say that people do care about things other than the value of their time, but they do so in a predictable way, so that their behavior can still be modeled and predicted. It is the job of researchers to ascertain what other things people care about.

Of course, users are not completely irrational; the first thing they care about is the absolute amount they will save. Thus, we propose:

*H1*: *To complete the purchase of a multi-item list, absolute savings will be positively correlated with the decision to go to another store. (Absolute Hypothesis)*

Despite this, we propose that other, predictably irrational, factors play a role in the decision to separate a basket of groceries into two trips.

## Relative value bias

Another finding of behavioral economics is relative value bias [23]. This bias is normally activated when deciding to go to a second vendor to realize savings on a particular product, thus trading time for money. Decision makers consider not only the absolute value that is to be saved, but also the relative savings in relation to the price of the item [23–26]. It has been shown that relative price discounts are significant in getting people to decide to purchase the item at the further location, even though the absolute savings are the same [23, 26]. Thus, we propose that relative value bias is present when making a multiple-item purchase:

*H2*: *To complete the purchase of a multi-item list, the relative value of savings for going to a second store will be positively correlated with the decision to go to another store. (Relative Hypothesis)*

## Quantity bias

The key theoretical addition we provide is to propose that in addition to the well-known biases discussed above, in multi-item basket shopping there will also be a quantity bias such that shoppers will prefer to go to a second store more if there are more items to purchase at the second store even when the relative and absolute savings are the same. This follows directly from Thaler's work on hedonic editing [13, 14]. Following Kahneman and Tversky (1981) Thaler (1985) posits a value function which is concave in gains, where consumers judge losses and gains based on some reference point and, most importantly, people can choose to aggregate or separate joint outcomes. Under these assumptions several results concerning joint outcomes can be shown. Of specific interests to this work is the idea that multiple gains should be segregated.

To see this, we start with the two-item case from Thaler (1985). First, we note that in the context of shopping savings is a gain relative to a base price. If a customer can get item 1 for price $X^*_1$ in one trip, but could get the same item for price $X_1$ by making two trips, where $X^*_1 > X_1$, then the customer enjoys a gain because the actual price paid is less than the established reference price. We label this gain $G_1$. If consumers have a value function which is concave over gains then $v(G_1) + v(G_2) > v(G_1 + G_2)$. By induction it is straightforward to show more generally that $\sum_{i=1}^{n} v(G_i) > v(\sum_{i=1}^{n} G_i)$. In other words, the value of saving a smaller amount on $n$ separate items is higher than the value of saving the sum of those amounts on a single

item. For hedonic editing the question is, should an individual aggregate gains? Whereas in our case the question is, would an individual prefer several small gains to a single equivalent large gain?

Thaler tested this with the following question. "Mr. A was given tickets to two lotteries involving a World Series. He won $50 in one lottery and $25 in the other. Mr. B was given a ticket to a single, larger World Series lottery. He won $75." [13, pg. 203]. When subjects were asked who would be happier, 56% said A would be happier, while only 16% said B would be happier and 15% said no difference. This result has been validated across many contexts including sales of stocks [27], manuscript acceptances [28], and weight loss during dieting [29]. Our contribution is to extend it to the case of N items and to use it not to predict how people should aggregate gains and losses, but rather to predict how people choose whether to avail themselves of saving allowed by new IT enabled comparison-shopping applications.

The notion of hedonic editing suggests that consumers would prefer savings to be spread across multiple items rather than to gain the same amount of savings on one or a few items. This suggests that consumers would be more willing to travel to a second store for savings if there are more items that will need to be purchased at the second store to achieve those savings. Thus, we predict:

*H3*: *To complete the purchase of a multi-item list, the number of items that are to be purchased from a second store will be positively correlated with the decision to go to the second store. (Quantity Hypothesis)*

## Methodology

We conducted three studies that confirm the existence of quantity bias. The first study involved a within-subject design, while studies 2 and 3 utilized between subject design. Our first study confirmed that quantity bias existed, while study 2 and 3 were utilized to explore properties about quantity bias. All studies were approved under IRB2018-763 by the Texas Tech University Human Research Protection Program, with participant's informed consent gained prior to starting each study.

### Methodology and data collection study 1

For our first study, we presented eight shopping situations, manipulating three variables between low and high values for a two by two-by-two presentation (see Table 1): total cart cost, potential total savings by going to a second store, and total number of items to be purchased at the second store. Total items in the cart was centered on 20 with a randomized presentation value of {19, 20, 21}. Total cart cost was centered on $50 and $100; with a randomized presentation value +/- 50 cents. Potential total savings by going to the second store was centered on $5 and $10; with a randomized presentation value +/- 50 cents. Total number of items to purchase at the second store were randomized between two sets: {1,2} and {9,10,11}. All stimuli, along with the total items, were randomized to reduce order effects from the repeated measures presentations. Additionally, cart order (the eight shopping situations) was also random.

Table 1. Anchor values for each cart presentation (Study 1).

| Variable | Low Value | High Value | +/- |
|----------|-----------|------------|-----|
| Cart Cost | $50.00 | $100.00 | $0.50 |
| Potential Savings | $5.00 | $10.00 | $0.50 |
| # of Items at Second Store | {1,2} | {9,10,11} | N/A |

**Table 2. Demographic data (n = 101) (Study 1).**

| Sex | n | % |
|---|---|---|
| Male | 43 | 42.60% |
| Female | 58 | 57.40% |
| Age (in years) | n | % |
| 18 | 7 | 7.00% |
| 19 | 5 | 5.00% |
| 20 | 12 | 12.00% |
| 21 | 36 | 36.00% |
| 22 | 25 | 25.00% |
| 23 | 10 | 10.00% |
| 24 | 1 | 1.00% |
| 25 | 1 | 1.00% |
| 26 | 1 | 1.00% |
| 29 | 1 | 1.00% |
| 34 | 1 | 1.00% |

All random values were uniformly randomized amongst the range or set, with total number of items randomly selected from {19, 20, 21} items.

Subjects were students at a large Southern university, who were asked to complete a survey about a new grocery shopping application. They received course credit for participating in the survey. A preliminary question that described the setup for the eight questions was presented which controlled for time by assuming that the second store would always be 10 minutes away. Then the eight different presentations were presented in random order. Likelihood for going to a second store was collected on a seven-level Likert scale.

A demographic summary of subjects is presented in Table 2.

After obtaining consent, demographic information was collected from each participant, and then a setup example question (Fig 2). Note that a math error exists in the setup question. The cheaper item was incorrectly multiplied by three at the second store instead of five. Thus, the example question shows a $.15 savings vs $.25 savings if going to the second store (no data was collected on the example question). We had purposely chosen a minimal amount close to zero, to avoid priming subjects for the follow-on presentations. Even with the example problem math error, we do not feel it affected participant responses in the experimental portion as the manipulated stimuli contained no errors in any of the presentations and therefore does not invalidate the results. Then eight questions were presented in the following format (Fig 3), in random order. All stimuli presentations were specific-product neutral so as to avoid any product-specific inherent biases from participants.

## Results study 1

The mean values for each cart type are presented in Table 3 (n = 101). The comparison values between each hypothesis are presented in Table 4. For H1 (absolute hypothesis) and H3 (quantity hypothesis), a paired t-test (n = 404) was performed to determine that the true difference in the means is not equal to 0. For H2 (relative hypothesis), a one-way ANOVA test was used to determine that there is a difference in the means between the three different conditions.

All three hypotheses are confirmed using paired t-tests and ANOVA. To test the robustness of the data to outliers, the results were re-analyzed by Winsorizing the data to discard the bottom and top five percent of respondents with regards to time completing the survey. All results remained significant.

The following sections will now ask you a question based on a hypothetical grocery list.

The items in your list are available at more than one store. The test application will always show you the cheapest store that you can fulfill your list, but there may be a chance for you to save money by buying some of your items at another store. You will assume that Store #2 is 10 minutes away.

For instance, macaroni and cheese might be $1.19 at Store #1 (and you need 5 of them) and bread is $2.30 (and you need 3 of them). At store #2, the macaroni and cheese is $1.14, and the bread is $2.47. Based on this you would be shown the following:

Total Items in Cart: 8
Total Cost at Store #1 is $12.85
If you also go to Store #2, save $.15
You would buy 3 items at Store #2

How likely are you to go to Store #2 to buy the 3 item(s)?

  ○ Extremely likely
  ○ Moderately likely
  ○ Slightly likely
  ○ Neither likely nor unlikely
  ○ Slightly unlikely
  ○ Moderately unlikely
  ○ Extremely unlikely

**Fig 2. Setup information provided to participants in study 1.**

In addition to the group-wise tests, we conducted regression analysis to determine coefficient size for each independent variable, including controls for age and sex. Regression analysis includes results for Ordered Logistic Regression (Table 5) and Linear Regression (Table 6) with a step-wise addition of each variable in our final model: Absolute Savings (by going to the 2$^{nd}$ store), Relative Savings (by going to the 2$^{nd}$ store), Number of Items to purchase at 2$^{nd}$ Store, Sex, and Age. R base version 3.4.2 was utilized with ordered logistic regression performed using the polr function of the MASS package version 7.3–47 and linear regression performed using the stats package of the base R version.

Again, all three hypotheses are confirmed and are significant in all models. Results were similar when running Winsorized regression against the removal of participants in the top and

Total Items in Cart: 19
Total Cost at Store #1 is $49.92
If you also go to Store #2, save $9.62
You would buy 1 item(s) at Store #2

How likely are you to go to Store #2 to buy the 1 item(s)?

  ○ Extremely likely
  ○ Moderately likely
  ○ Slightly likely
  ○ Neither likely nor unlikely
  ○ Slightly unlikely
  ○ Moderately unlikely
  ○ Extremely unlikely

**Fig 3. Example study 1 question.**

**Table 3. Mean cart values (n = 101) (Study 1).**

| Avg Cart Cost | Absolute Savings by going to 2nd store | Items to purchase at 2nd Store | Mean Response (Likert 1–7) |
|---|---|---|---|
| $50 | $5 | 1 or 2 | 3.25 |
| $50 | $5 | 9, 10, or 11 | 3.57 |
| $50 | $10 | 1 or 2 | 4.73 |
| $50 | $10 | 9, 10, or 11 | 5.08 |
| $100 | $5 | 1 or 2 | 2.81 |
| $100 | $5 | 9, 10, or 11 | 3.01 |
| $100 | $10 | 1 or 2 | 4.23 |
| $100 | $10 | 9, 10, or 11 | 4.72 |

Likert Scale Values: Extremely unlikely (1), Moderately unlikely (2), Slightly unlikely (3), Neither likely nor unlikely (4), Slightly likely (5), Moderately likely (6), Extremely likely (7).

bottom five percent with regards to time to complete the survey. Effect size classifies as small for quantity bias (r = .08), small for relative value bias (r = .11), and medium to large for absolute savings (r = .39) [30]. Of note, for both types of regression analysis in the full model, age and sex are also significant, decreasing the likelihood of going to another store to complete a multi-item purchase as age increases or if the respondent is male.

## Relative vs. absolute quantity bias (Study 2)

After verifying the existence of quantity bias, we wanted to see if the bias is relative or absolute, i.e. does it matter how many items are at a second store when compared to the total number of items? Since relative value bias has shown to be an influence on the perception of the value of absolute savings, it is worth testing to see if quantity bias also has a relative component. It is possible that, if we fix the number of items at the second store, and increase the total number of items systematically, there could be a systematic change in the willingness to go to the second store. Thus, we propose:

H4: *Willingness to travel to the second store will increase as the relative number of items that are less expensive at the second store decreases. (Relative-Quantity Hypothesis)*

**Table 4. Hypothesis results (paired t-test and ANOVA) (Study 1).**

| Hypothesis | Condition 1 | Condition 2 | Condition 3 |
|---|---|---|---|
| H1. Absolute savings value will be positively correlated with the choice to shop at the 2nd store. | Savings = $5 (n = 404) | Savings = $10 (n = 404) | N/A |
| Mean response | 3.16** | 4.57** | N/A |
| H2. Relative savings value will be positively correlated with the choice to shop at the 2nd store. | Savings = 5% (n = 202) | Savings = 10% (n = 404) | Savings = 20% (n = 202) |
| Mean response | 2.91** | 3.94** | 4.91** |
| H3. The number of items to purchase at the 2nd store will be positively correlated with the choice to shop at the 2nd store. | Items = 1 or 2 (n = 404) | Items = 9, 10, or 11 (n = 404) | N/A |
| Mean response | 3.75** | 4.1** | N/A |

P-Value Significance Codes

'**' < 0.01

'*' < 0.05

'†' < 0.1.

**Table 5. Ordered Logistic Regression results against likelihood (n = 808) (Study 1).**

| Independent Variable | Model | | | | |
|---|---|---|---|---|---|
| | AS | AS + RS | AS + RS + Items | AS + RS + Items + Sex | AS + RS + Items + Sex + Age |
| | Value | Value | Value | Value | Value |
| Absolute Savings (AS) | 0.298** | 0.217** | 0.218** | 0.218** | 0.223** |
| Relative Savings (RS) | | 0.057** | 0.057** | 0.057** | 0.058** |
| # Items at 2nd Store (Items) | | | 0.036** | 0.036** | 0.038** |
| Sex (male) | | | | -0.223† | -0.327* |
| Age | | | | | -0.147** |

P-Value Significance Codes

'**' < 0.01

'*' < 0.05

'†' < 0.1.

P-Values computed by comparing t-values against standard normal distribution.

**Methodology and data collection study 2.** We also test again for quantity bias (H3 from above).

In addition, to testing for a relative effect of quantity bias, we also wanted to expand our sample beyond college students to increase generalizability. Therefore, we collected a sample using Amazon Mechanical Turk by advertising a 20-cent Human Intelligence Task (HIT) asking for 1–2 minutes to answer questions about grocery shopping decisions and two demographic questions (gender and age). A demographic summary of the final subject pool is presented in Table 7. The sample is much more varied in age than the college student sample.

We presented two scenarios to 398 participants. One scenario where the total number of items varied uniformly from 2 to 21, with a single item cheaper at the second store. The other scenario where the total number of items varied uniformly from 6 to 25, with 5 items cheaper at the second store. Held constant was the time to the second store (5 minutes), cost of all items at the first store ($50), and the savings by buying the cheaper items at second store ($5). To make sure subjects read the question we required them to repeat back each of the variables

**Table 6. Linear Regression results against likelihood (n = 808) (Study 1).**

| Independent Variable | Model | | | | |
|---|---|---|---|---|---|
| | AS | AS + RS | AS + RS + Items | AS + RS + Items + Sex | AS + RS + Items + Sex + Age |
| | Value | Value | Value | Value | Value |
| Intercept | 1.669** | 1.659** | 1.444** | 1.540** | 4.433** |
| Absolute Savings (AS) | 0.300** | 0.219** | 0.219** | 0.219** | 0.220** |
| Relative Savings (RS) | | 0.055** | 0.055** | 0.055** | 0.055** |
| # Items at 2nd Store (Items) | | | 0.038** | 0.038** | 0.038** |
| Sex (male) | | | | -0.226† | -0.298* |
| Age | | | | | -.0134** |
| Adjusted R2 | 0.156 | 0.168 | 0.174 | 0.176 | 0.196 |

P-Value Significance Codes

'**' < 0.01

'*' < 0.05

'†' < 0.1 Note: When running a fixed effect model, sex and age are no longer significant, however, all remaining variables become much more significant. We present the random effects model to err on the side of caution.

**Table 7. Demographic data (n = 371) (Study 2).**

| Sex | n | % |
|---|---|---|
| Male | 175 | 47% |
| Female | 196 | 53% |
| Other | 0 | 0% |
| **Age (in years)** | **n** | **%** |
| 18–20 | 27 | 7.30% |
| 21–30 | 188 | 50.70% |
| 31–40 | 85 | 22.90% |
| 41–50 | 45 | 12.10% |
| 51–60 | 21 | 5.70% |
| 61–70 | 4 | 1.10% |

first, then we asked them how likely they were to go to the second store (see Fig 4). Each participant received one scenario presentation. After removing responses in which respondents did not repeat back at least four of the five variables correctly, 371 valid responses were analyzed.

## Results study 2

An overview of the presentation values is shown in Table 8.

You need to buy some groceries. You can either buy all of the groceries at one store, or split your list and go to two stores. If you split your grocery list and go to two stores you can save some money. The details of each option are below.
Extra driving time if you go to the second store: 5 minutes
Total number of items on grocery list: 20
Cost of all groceries at one store: $50
Savings if you go to the second store: $5
Number of items that you could get cheaper at the second store: 1

How many minutes away is the second store?

How much are all your items at the first store?

How many total items are you buying?

How much do you save by going to the second store?

How many items would you buy at the second store?

How likely are you to go to the second store to buy the 1 item?

- Extremely likely
- Moderately likely
- Slightly likely
- Neither likely nor unlikely
- Slightly unlikely
- Moderately unlikely
- Extremely unlikely

**Fig 4. Example study 2 question.**

**Table 8. Presentation values (n = 371) (Study 2).**

| # Items Cheaper at Second Store | Total Items | Mean Likelihood | n |
|---|---|---|---|
| 1 | {2–21} | 3.08 | 186 |
| 5 | {6–25} | 4.15 | 185 |

Total cost was fixed at $50, savings at $5, time to second store at 5 minutes. Participant responses were those that correctly answered at least 4 of 5 check questions.

We conducted regression analysis of the likelihood of shopping at a second store against the total number of items in the basket (relative-quantity effect) with a dummy variable indicating whether the total number of items purchased at the second store was 1 or 5 (the absolute-quantity effect) and we include controls for age and sex. Regression analysis included Ordered Logistic Regression and Linear Regression with a step-wise addition of each variable in our final model: Total Items at Second Store, Gender, and Age. R base version 3.4.2 was utilized with ordered logistic regression performed using the polr function of the MASS package version 7.3–47 and linear regression performed using the stats package of the base R version.

Regression against the total number of items in the baskets was not significant for any model, while # of items at the second store was significant. Thus, H3 (quantity bias) was supported and H4 (relative-quantity) was not supported. (Note, tables not included for space, but are available if desired).

We then took a subset of each presentation value such that the total number of items was limited to those that were between 5 and 22 (see Table 9) to investigate presentations where the total number of items were included on both values of the control condition, so as to determine if H4 (relative-quantity) remained insignificant on the overlap. Then we ran regression against the likelihood of shopping at a second store against the number of cheaper items at the second store and total number of items, including controls for age and gender. Regression analysis included Ordered Logistic Regression (Table 10) and Linear Regression (Table 11) with a step-wise addition of each variable in our final model: Total Items at Second Store, Gender, and Age. R base version 3.4.2 was utilized with ordered logistic regression performed using the polr function of the MASS package version 7.3–47 and linear regression performed using the stats package of the base R.

*H3 (Quantity Bias) is supported in all models. However, H4 (Relative-Quantity) is not supported in this analysis either. Effect size for quantity bias is medium (r = .28) [30]. Within this study age and gender are not significant.*

## Online vs. offline effects (Study 3)

Thus far our focus has been on examining crowdsourced price comparison for physical trips to the grocery store. However, quantity bias may also exist for online shopping. Though there is not a physical cost to travel to a second online retailer, there is a cost in terms of time and effort to visit two different online shopping channels. Thus, in study 3 we test to see if the quantity bias exists for online shopping as well. As we argue above, quantity bias is based on the general rule that more items lead to a greater total perceived value for a shopping trip. We believe this inherent bias does not simply vanish in an online setting because the same logic applies, and the same general rule can be derived. Currently, the authors are unaware of any multi-item price comparison websites that are as in-depth as the applications presented for physical store shopping. However, it is not difficult to understand that it is technologically feasible to apply these same design characteristics to online-comparisons of multi-item baskets, especially as online-grocery shopping becomes more commonplace.

**Table 9. Presentation values (n = 371) (Study 2).**

| 1 item at second store | | | 5 items at second store | | |
|---|---|---|---|---|---|
| **Total Items** | **Mean Likelihood** | **n** | **Total Items** | **Mean Likelihood** | **n** |
| {2–21} | 2.9 | 186 | {6–25} | 4.17 | 185 |
| 2 | 3.56 | 9 | 6 | 5.2 | 10 |
| 3 | 3.64 | 11 | 7 | 3.62 | 8 |
| 4 | 4.2 | 10 | 8 | 4 | 8 |
| 5 | 3.67 | 9 | 9 | 3.7 | 10 |
| 6 | 2.17 | 6 | 10 | 4.7 | 10 |
| 7 | 3.33 | 9 | 11 | 2.9 | 10 |
| 8 | 2.7 | 10 | 12 | 4.27 | 11 |
| 9 | 3.3 | 10 | 13 | 4 | 6 |
| 10 | 1.9 | 10 | 14 | 3.64 | 11 |
| 11 | 3 | 11 | 15 | 3.8 | 10 |
| 12 | 2.44 | 9 | 16 | 4 | 9 |
| 13 | 3.11 | 9 | 17 | 4.44 | 9 |
| 14 | 2.67 | 9 | 18 | 4.67 | 9 |
| 15 | 4.22 | 9 | 19 | 4.78 | 9 |
| 16 | 3.89 | 9 | 20 | 3.88 | 8 |
| 17 | 1.25 | 8 | 21 | 5.25 | 8 |
| 18 | 2.33 | 9 | 22 | 3.8 | 10 |
| 19 | 3.33 | 9 | 23 | 4.3 | 10 |
| 20 | 2.4 | 10 | 24 | 4 | 9 |
| 21 | 3.9 | 10 | 25 | 4.1 | 10 |

Total cost was fixed at $50, savings at $5, time to second store at 5 minutes. Participant responses were those that correctly answered at least three of five check questions.

The decision to make the trip then is a comparison of the value of the trip to the cost of the trip. As online shopping is generally less time consuming than physical store shopping, we expect that the threshold that the value of the trip must overcome is lower for online shopping.

If we are correct, this would result in a main effect of both quantity and shopping method, but a non-significant interaction. Thus, we hypothesize:

**Table 10. Ordered logistic results against likelihood (n = 293).**

| Independent Variable | Model | | | |
|---|---|---|---|---|
| | **2nd** | **2nd + total** | **2nd + total + male** | **2nd + total + male + Age** |
| | **Value** | **Value** | **Value** | **Value** |
| # of Items at second store = 5 (2nd) | 1.100** | 1.112** | 1.110** | 1.113** |
| Total # of items (total) | | 0.026 | 0.027 | 0.026 |
| Gender male (male) | | | 0.148 | 0.154 |
| Age | | | | 0.006 |

P-Value Significance Codes

'**' < 0.01

'*' < 0.05

'†' < 0.1. P-Values computed by comparing t-values against standard normal distribution. Note: results run on participants that correctly answered at least four of five check questions. The analysis was also run on participants that correctly answered at least three of five and five of five correctly, with similar results.

**Table 11. Linear Regression results against likelihood (n = 293) (Study 2).**

| Independent Variable | Model | | | |
|---|---|---|---|---|
| | 2nd | 2nd + total | 2nd + total + male | 2nd + total + male + Age |
| | Value | Value | Value | Value |
| Intercept | 2.898** | 2.538** | 2.422** | 1.975** |
| # of Items at second store = 5 (2nd) | 1.273** | 1.280** | 1.276** | 1.300** |
| Total # of items (total) | | 0.026 | 0.027 | 0.027 |
| Gender male (male) | | | 0.227 | 0.24 |
| Age | | | | 0.014 |
| Adjusted R2 | 0.08 | 0.08 | 0.08 | 0.081 |

P-Value Significance Codes

'**' < 0.01

'*' < 0.05

'†' < 0.1. P-Values computed by comparing t-values against standard normal distribution. Note: results run on participants that correctly answered at least four of five check questions. The analysis was also conducted on participants that correctly answered at least three of five and five of five correctly, with similar results.

H5: *To complete the purchase of a multi-item list, there is a positive effect of online shopping on willingness to go to a second store. (Quantity-Online)*

H6: *There is no significant interaction between online vs. offline shopping and number of items. (Quantity-Online Offline-Items)*

## Methodology and data collection study 3

We presented a two-by-two matrix, manipulating the number of items at the second store between one and 10, and the purchase location between online and offline. We kept the total number of items, total cost, and savings, constant at 20 items, $50, and $5 respectively. We also kept time to the second store for offline presentation and time to complete transaction on the other website the same at five minutes (see Table 12).

Subjects were collected using Amazon Mechanical Turk by advertising a 20-cent Human Intelligence Task (HIT) asking for 1–2 minutes of their time about their shopping habits. Two demographic questions (gender and age), five check questions, and one of the four scenario situations were presented, with likelihood for going to a second store or online retailer collected on a seven-level Likert scale. Subjects from the previous study were excluded as participants in this study, and subjects in this study were only allowed to participate once. (see Fig 5).

A demographic summary of subjects is presented in Table 13.

## Results study 3

In this study, we collected responses from 496 participants on Amazon Mechanical Turk. Again, we only use participants that got at least four of five check questions correct. This left

**Table 12. Anchor values for each cart presentation (Study 3).**

| Variable | Low Value | High Value |
|---|---|---|
| Purchase environment | online | offline |
| # of Items at Second Store | 1 | 10 |

One presentation value for each participant. Total cost of $50, time to second store or extra transaction time at online retailer of five minutes, and a savings of $5 by going to the second store were fixed for each presentation.

You need to buy some items online. You can either buy all of the items at one online retailer, or split your list and use two online retailers. If you split your list and use two online retailers, you can save some money. The details of each option are below.
Extra transaction time if you also purchase from the second online retailer: 5 minutes
Total number of items to purchase: 10
Cost of all items at the first online retailer: $50
Savings if you also purchase at the second online retailer: $5
Number of items that you could get cheaper at the second online retailer: 5

How many extra minutes if you use the second online retailer?

How much are all your items at the first online retailer?

How many total items are you buying?

How much do you save by also purchasing at the second online retailer?

How many items would you buy at the second online retailer?

How likely are you to use the second online retailer to buy the 5 items?

- ○ Extremely likely
- ○ Moderately likely
- ○ Slightly likely
- ○ Neither likely nor unlikely
- ○ Slightly unlikely
- ○ Moderately unlikely
- ○ Extremely unlikely

**Fig 5. Example study 3 question.** (Note, offline questions were identical in setup as Fig 4, except updated to reflect 10 total items).

n = 464. Results were re-run against at least three of five, and five of five check questions with similar results.

We conducted regression analysis to determine the coefficient size for each presentation, including controls for age and sex. Regression analysis included Ordered Logistic Regression

**Table 13. Demographic data (n = 464) (Study 3).**

| Sex | n | % |
|---|---|---|
| Male | 193 | 43.30% |
| Female | 268 | 60.10% |
| Other | 2 | <0.01% |
| No Answer | 1 | <0.01% |
| **Age (in years)** | **n** | **%** |
| 18–20 | 20 | 4.50% |
| 21–30 | 250 | 56.10% |
| 31–40 | 123 | 27.60% |
| 41–50 | 43 | 9.60% |
| 51–60 | 21 | 4.70% |
| 61–70 | 7 | 1.60% |

**Table 14. Ordered logistic results against likelihood (n = 464) (Study 3).**

| Independent Variable | Model | | | | |
|---|---|---|---|---|---|
| | % | % + online | % + online + online\*# | % + online + online\*# + male + other + NA | % + online + online\*# + male + other + NA + Age |
| | Value | Value | Value | Value | Value |
| # of items at second store (#) | 0.135** | 0.153** | 0.182** | 0.176** | 0.175** |
| Online (online) | | 1.746** | 1.924** | 1.911** | 1.900** |
| Interaction of Online * # (online*#) | | | -0.058 | -0.057 | -0.055 |
| Gender male (male) | | | | 0.042 | 0.049 |
| Gender other (other) | | | | -1.875 | -1.836 |
| Gender not indicated (NA) | | | | 13.303 | 13.314 |
| Age | | | | | 0.005 |

P-Value Significance Codes

'**' < 0.01

'*' < 0.05

'†' < 0.1. P-Values computed by comparing t-values against standard normal distribution. Note: results run on participants that correctly answered at least four of five check questions. The analysis was also conducted on participants that correctly answered at least three of five and five of five correctly, with similar results.

(Table 14) and Linear Regression (Table 15) with a step-wise addition of each variable in our final model: # of items at the second store, a dummy variable for Online vs. Offline presentation (Online), Gender, and Age. R base version 3.4.2 was utilized with ordered logistic regression performed using the polr function of the MASS package version 7.3–47 and linear regression performed using the stats package of the base R version.

Both types of regression analysis support H3 (Quantity Bias), H5 (Quantity-Online), and H6 (Quantity-Online Offline-Items) with controls Age and Gender not becoming significant

**Table 15. Linear Regression results against likelihood (n = 464) (Study 3).**

| Independent Variable | Model | | | | |
|---|---|---|---|---|---|
| | % | % + online | % + online + online\*# | % + online + online\*# + male + other + NA | % + online + online\*# + male + other + NA + Age |
| | Value | Value | Value | Value | Value |
| Intercept | 4.058** | 3.060** | 3.004** | 2.99** | 2.913** |
| # of items at second store (#) | 0.173** | 0.179** | 0.197** | 0.188** | 0.187** |
| Online (online) | | 1.976** | 2.09** | 2.08** | 2.075** |
| Interaction of Online * # (online*#) | | | -0.036 | -0.036 | -0.035 |
| Gender male (male) | | | | 0.127 | 0.13 |
| Gender other (other) | | | | -2.7 | -2.68 |
| Gender not indicated (NA) | | | | 1.168 | 1.177 |
| Age | | | | | 0.003 |
| Adjusted R2 | 0.022 | 0.216 | 0.2145 | 0.217 | 0.216 |

P-Value Significance Codes

'**' < 0.01

'*' < 0.05

'†' < 0.1. P-Values computed by comparing t-values against standard normal distribution. Note: results run on participants that correctly answered at least four of five check questions. The analysis was also run on participants that correctly answered at least three of five and five of five correctly, with similar results.

as in Study 1. Quantity Bias exists for both offline and online shopping at roughly the same level. Effect size for quantity bias is small to medium (r = .15) [30].

## Discussion and conclusion

Prior research has demonstrated that people are predictably irrational in that they attend to the relative cost savings rather than only attending to the absolute cost savings when making a decision to travel to a store. We add to the literature on predicable irrationality in the decision to travel to another store by proposing that the number of items will also be a significant decision variable for people making a decision about traveling to a store. We term this quantity bias. We test this theory in a series of three studies and consistently produce the result across populations and methods.

First, using college students and a within-subjects design, we demonstrate that the bias exists even after controlling for absolute and relative savings. Second, using a sample from Mechanical Turk and a between-subjects design, we confirm quantity bias. Third, using a different sample from Mechanical Turk, we demonstrate that quantity bias exists both in online and physical shopping. Thus, we find quantity bias in three different samples under three different conditions. Effect size for all three studies ranges from small to medium, which indicates that while not as powerful an effect as absolute savings, which has a medium to large effect size in study 1, quantity bias does have an impact on user decisions to go to a further on store, or to make a purchase online at a different retailer.

We also note that there are direct consequences for the designers of comparison-shopping applications that directly influence utility, i.e. the intensity of pleasure or pain that is gained from actions [31]. Including information about the number of items seems to influence consumers, so designers need to proceed carefully when choosing how to display such information. In principle, consumers could be nudged into making better decisions if the information is excluded. However, this strategy could backfire if consumers actually feel bad about buying a single item in one trip. It would be useful to do further experiments to determine if experienced and remembered utility were compromised by having subjects only purchase one item. Thus far we have only demonstrated that anticipated utility is compromised. It is easy to imagine a situation where a comparison-shopping app failed because people complained that it made them take an extra trip for only a few items. However, it is also easy to imagine the same app succeeding because people saved more by not being presented with information that activated predictably irrational behavior. More research is warranted.

There could also be implications that sellers need to consider as these sorts of physical world comparison-shopping applications become more widely adopted. The first is that it does not seem that customers will completely price optimize even after considering travel costs. Thus, strategies like loss leaders [32, 33] where a retailer offers an item at a very low cost to get customers into a store to buy other items at regular prices could endure, but the effect of price comparison apps may induce more negative results.

Another interesting thing that we discovered was that while people still experienced quantity bias to a similar degree online, there was a much higher baseline willingness to go to a second store online. This occurs in spite of the fact that in both cases we suggested that the time involved was identical in the online and offline contexts. It might be the case that in this work we unintentionally discovered a second economic decision bias that makes people discount time spent online at a different rate than time spent in the physical world. More research would be required to assess this. However, the immediate implication of our inquiry is that online retailers might be much more susceptible to comparison-shopping even for one or few items.

Research supporting offline time-denominated mental accounts has been reported [34] while other research has not found the same [25, 35]. Additionally, some research has shown that people are more price sensitive online than offline, especially when cross-store comparison is made easy [36], while other research has shown lower price sensitivity online [36, 37] when differentiated product information is highlighted. Research into the predicable irrationality of both sensitivities and time-denominated mental-accounts is worthy of more research when considering quantity bias. We should also note that there is an alternative rational explanation for quantity bias. If we assume that a store could be out of an item, then we would have to discount the savings from items by the probability of the item being out of stock. In other words, going to a second store would be a gamble between getting a savings and getting nothing if the store is out of stock. For a single item, this would be a particularly risky gamble. Of course, this becomes very complex for multi-item shopping, so more research should be done. However, it does suggest that in addition to costs, applications might want to offer information on quantity or availability of items.

In conclusion, we note that quantity bias seems to be a robust effect, which we were able to duplicate in multiple settings. It is a heretofore unexamined bias that has many important consequences for the design of comparison-shopping systems. It seems to have differential consequences for online vs. offline sellers, so there is an important technology component that needs to be studied. It also has implications for how researchers can model and predict both shopper behavior and higher-level market trends in the presences of comparison-shopping applications. In short, it seems to be a pretty important consideration for both researchers and practitioners, which demands additional study.

## Author Contributions

**Data curation:** Ross Niswanger.

**Formal analysis:** Ross Niswanger, Eric Walden.

**Investigation:** Ross Niswanger, Eric Walden.

**Methodology:** Ross Niswanger, Eric Walden.

**Project administration:** Eric Walden.

**Validation:** Ross Niswanger.

**Writing – original draft:** Ross Niswanger, Eric Walden.

**Writing – review & editing:** Ross Niswanger, Eric Walden.

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
