## [Decision Letter · Decision Letter 0]

24 Nov 2021

PONE-D-21-23401

Quantity Bias in comparison-shopping of multi-item baskets

PLOS ONE

Dear Dr. Niswanger,

Thank you for submitting your manuscript to PLOS ONE. After careful consideration, we feel that it has merit but does not fully meet PLOS ONE’s publication criteria as it currently stands. Therefore, we invite you to submit a revised version of the manuscript that addresses the points raised during the review process.

We look forward to receiving your revised manuscript.

Kind regards,

James H. Cardon, Ph.D.

Academic Editor

PLOS ONE

Additional Editor Comments (if provided):

Dear Darren,

I have now received 2 thoughtful reviews of your paper, both writing that the paper is nice contribution to the literature on behavioral economics. The paper in its present form is not ready for publication, so I ask that you revise the paper, paying ca close attention to the comments and suggestions. Both reviewers noticed a possible error in Study 1, but there other points that need clarification.

I look forward to reading your revision. Thank you for submitting to PLOS ONE.

James Cardon

Journal Requirements:

4. We note that Figure 1 in your submission contain copyrighted images. All PLOS content is published under the Creative Commons Attribution License (CC BY 4.0), which means that the manuscript, images, and Supporting Information files will be freely available online, and any third party is permitted to access, download, copy, distribute, and use these materials in any way, even commercially, with proper attribution. For more information, see our copyright guidelines: http://journals.plos.org/plosone/s/licenses-and-copyright.

Reviewers' comments:

Reviewer's Responses to Questions

**Comments to the Author**

1. Is the manuscript technically sound, and do the data support the conclusions?

Reviewer #1: Yes

Reviewer #2: Partly

2. Has the statistical analysis been performed appropriately and rigorously? 

Reviewer #1: Yes

Reviewer #2: Yes

3. Have the authors made all data underlying the findings in their manuscript fully available?

Reviewer #1: Yes

Reviewer #2: No

4. Is the manuscript presented in an intelligible fashion and written in standard English?

Reviewer #1: Yes

Reviewer #2: Yes

5. Review Comments to the Author

Reviewer #1: Quantity Bias in comparison-shopping of multi-item baskets

Manuscript: PONE-D-21-23401

This paper proposes and examines a new behavioral bias, quantity bias, in which consumer decisions concerning travel-to-save trade-offs are influenced, in a predictably irrational way, by the number of items on which a saving can be made, even after controlling for absolute and relative saving levels.

The motivation and development of the argument in support of a quantity bias and the associated hypotheses is well-grounded in prior literature and clearly explained. The empirical analyses undertaken are suitable to test the hypotheses and the results are clear and suitably explained. Across three studies, the paper provides evidence in support of the quantity bias. The findings have important implications for understanding consumer behavior.

I like the paper and believe it makes an interesting and important contribution to the behavioral economics and consumer decision literatures. The experimental design is novel and offers unique insight not available in other experiments examining travel-to-save trade-offs. As such, the paper has strong publication potential. Below I provide a number of suggestions on how the paper might be improved prior to publication.

Motivation and hypotheses

The motivation is clear and well-founded in prior literature, so I would not see much need for change here.

The only suggestion would be to broaden the discussion around the relative saving bias (mental accounting effect in the parlance of Kahneman, Tversky, Thaler, among others). The explanation for the relative saving effect offered in the mental accounting literature centers on the how individuals construct and use mental accounts. If they form topical accounts (or what might be termed item accounts in the parlance of the current study), they compare the value of the saving to the cost of the discounted item rather than the cost of the basket as would be the cases under comprehensive accounts (presumably leading to rational behavior). To examine the quantity bias necessitates an experimental design in which multiple items are discounted. This might be expected to weaken mental accounting effects unless narrow bracketing is sufficiently strong. While the explanation in the paper based on hedonic editing and the associated segregation of gains (i.e., savings) is convincing as to why more discounted items might lead to a higher likelihood of travel to the second store, it might be helpful to the reader to relate the discussion back to the notion of topical mental accounts to further reinforce the idea of narrow bracketing.

Study 1

The experimental design and method are well-thought out and fit for purpose. There is slight confusion initially, however, when the design is presented as a 2x2x2 design, but the text then lists manipulations for four factors; total items, total cost, total savings, number of cheaper items at second store. Total items is not included as a factor in the discussion of the 2x2x2 design. What is its role in the experiment and how might it influence the role of the number of cheaper items at the second store? This is not clear at this point in the text and should be clarified.

There seems to be an error in the setup information presented in Figure 2. The macaroni and cheese product is cheaper at Store #2 then Store #1 ($1.14 vs $1.19, respectively) and 5 are needed, giving total saving of $0.25. But the Store #2 information indicates a total saving of $.15 based on buying 3 items. Presumably this is just a typo in the transposition from the experimental instrument to the paper, but this need to be checked. If it turns out to be an error in the instrument, the authors should consider if this is material to their findings and if so re-run the experiment (perhaps based on a reduced design and sample) to verify that the results still hold.

The three hypotheses are initially tested using 2 x paired t-test and one-way ANOVA. Given the within-subject, full-factorial design, a repeated measures multivariate ANOVA would seem suitable to test all three hypotheses and would also allow examination of potential interaction effects. Have the authors considered this? Interaction effects might also be added to the regression models. The potential interaction between total items and the number of cheaper items at the second store might be interesting to examine and could feed into Study 2.

Study 2

The decision to restrict analysis to a subset of the data would benefit from a fuller account and justification. The importance of why the total number of items is restricted between 5 and 22 could be clearer.

Study 3

No suggestions.

Discussion and Conclusion

The concluding section is informative and well-written, but could be strengthened in a number of ways.

First, there are multiple references to how “models” of consumer behavior need to take account of the quantity bias. It would be informative to refer explicitly to specific models here and include citations to relevant papers. Also, I am not unclear about the statement that such models should include a “distaste for single item shopping”. This could be construed to mean consumers prefer to buy multiple items in a single shopping trip rather than single items across multiple trips – something akin to weekly versus daily grocery shopping trips, which the paper does not examine.

Second, reference to experienced, remembered and anticipated utility is a little vague and would benefit from definition of these terms and citations to relevant papers.

Third, the possibility that people might discount time differently online vs physical world is an interesting. That individuals might be irrational in their valuation/use of their own time would benefit from linking to prior literature examining mental accounting in the context of time-denominated accounts, as opposed to money-denominated accounts. For example, when savings are in time rather than money, Leclerc et al. (1995) find evidence supporting the use of topical accounts, while findings in Frisch (1993) and Duxbury et al. (2005) suggest an absence of mental accounting effects for time.

More generally, in all studies, the rationale offered for the regression analyses is to determine coefficient size. While these are reported in the tables, it would help the reader’s interpretation if the paper included brief discussion of the effect sizes for the behavioral biases across the three studies.

Minor comments

1. On p5, “$28.0” in the main text should be “$28.01”.

2. Final sentence of first para on p.6 states researchers “determine” what consumers care about. But “determine” could be read as prescribe or dictate here, and I would recommend replacing by something more neutral such as “examine” or “evaluate”.

3. Also on p.6, the reference to “single-item decision” in the context of the cited studies of mental accounting effects is not quite precise. Such studies use scenarios in which two items are to be purchased at a single store (in contrast to the possibility in the current paper that items will be purchased from multiple stores), as per the classic jacket and calculator problem on p.3, but where one is discounted at a second store. The non-discounted price of the original item is varied to manipulate the relative saving while holding absolute saving constant. This is necessary to isolate the separate effects of absolute and relative savings. Such a design holds constant the value of the absolute saving relative to the total combined expenditure, so there is not rational economic reason for differences in likelihood of travel. Hence, the phrasing in the paper needs to be more precise here.

4. On p.7, the closing speech marks on the quote from Thaler should not enclose the in-text citation to Thaler’s work.

5. There are occasional sentences where I would have liked to see more use of commas, but this is sometimes a matter of personal preference.

6. On p.22, first para, on first reading the phrase “greater total value of the shopping trip” might easily be misconstrued as referring to total cost of the basket, which renders the sentence somewhat meaningless. On closer consideration, I take the phrase to be referring to the value or utility derived from the savings to be made from traveling to the second store. Is this correct? Either way, the sentence needs rephrasing to avoid potential misinterpretation.

7. In the reference for Mowen & Mowen (1986) the “*” can be removed from the title.

References

Duxbury, D., Keasey, K., Zhang, H., & Chow, S. L. (2005). Mental accounting and decision making: Evidence under reverse conditions where money is spent for time saved. Journal of Economic Psychology, 26(4), 567-580.

Frisch, D. (1993). Reasons for framing effects. Organizational Behavior and Human Decision Processes, 54(3), 399-429.

Leclerc, F., Schmitt, B. H., & Dube, L. (1995). Waiting time and decision making: Is time like money?. Journal of Consumer Research, 22(1), 110-119.

Reviewer #2: Dear authors:

This paper defines a new behavioral bias in consumer behavior, named “quantity bias”, and

develops three studies to test its existence and measure its effect. Quantity bias means that

people value more the same savings (absolute and relative) when they spread among more

items.

I like the paper. It is relevant and original. The methodology is appropriate to test and measure

the effects. However there are some issues that should be addressed before publication.

• If I am not wrong, Study 1 (Figure 2) has two errors: the shopper would buy 5 items

(instead of 3) at store 2 (macaroni and cheese is less expensive at store 2) and the

savings would be 0.25 dollars (instead of 0.15 dollars)

• Figure 3. I don’t understand the numbers. To buy 19 items in total, I think the only

possibility is that the shopper buys 5x2=10 macaroni and pasta and 3x3=9 bread items

(there is not other way to get 19 as factor of 5 and 3). Then the shopper would spend

32.6 dollars in store 1. To save going to store 2 the shopper would buy the bread at

this store (bread is cheaper at store 2), but in this case the shopper would buy 3x3=9

items of bread at stores 2, not 1 item. May be I am wrong.

• How many presentations were shown to each subject? 4? I say this because in page 9

it is said 8 presentations.

• Study 2. According to hedonic editing, the same saving is more valued when separated

in more items. Then, H4 should be rejected, isn’t it?

• P19: You say: “We conducted regression analysis of the likelihood of shopping at a

second store against the total number of items in the basket (relative effect)”

Shouldn’t the relative effect be instead # items bought at store 2 divided by # total

items bought?

• P19: You say: “Regression against the total number of items in the baskets was not

significant for any model, while # of items at the second store was significant. Thus, H3

(Quantity Bias) and H4 (Relative-Quantity) were not supported.” If # of items at the

second store was significant, doesn’t it imply that H3 is supported?

• Study 3 is for the online setting, and in my opinion is less relevant

6. PLOS authors have the option to publish the peer review history of their article (what does this mean?). If published, this will include your full peer review and any attached files.

Reviewer #1: No

Reviewer #2: No

---

## [Author Response · Author response to Decision Letter 0]

5 Jan 2022

Response to reviewers

Dear reviewers, thank you for your insightful comments. Below are our responses to each of your comments.

Reviewer #1

The only suggestion would be to broaden the discussion around the relative saving bias (mental accounting effect in the parlance of Kahneman, Tversky, Thaler, among others). The explanation for the relative saving effect offered in the mental accounting literature centers on the how individuals construct and use mental accounts. If they form topical accounts (or what might be termed item accounts in the parlance of the current study), they compare the value of the saving to the cost of the discounted item rather than the cost of the basket as would be the cases under comprehensive accounts (presumably leading to rational behavior). To examine the quantity bias necessitates an experimental design in which multiple items are discounted. This might be expected to weaken mental accounting effects unless narrow bracketing is sufficiently strong. While the explanation in the paper based on hedonic editing and the associated segregation of gains (i.e., savings) is convincing as to why more discounted items might lead to a higher likelihood of travel to the second store, it might be helpful to the reader to relate the discussion back to the notion of topical mental accounts to further reinforce the idea of narrow bracketing.

RESPONSE: We have expanded the discussion as suggested.

There seems to be an error in the setup information presented in Figure 2. The macaroni and cheese product is cheaper at Store #2 then Store #1 ($1.14 vs $1.19, respectively) and 5 are needed, giving total saving of $0.25. But the Store #2 information indicates a total saving of $.15 based on buying 3 items. Presumably this is just a typo in the transposition from the experimental instrument to the paper, but this need to be checked. If it turns out to be an error in the instrument, the authors should consider if this is material to their findings and if so re-run the experiment (perhaps based on a reduced design and sample) to verify that the results still hold.

RESPONSE: We have added language that talks to the situation that the setup problem did contain a math error, but that all presentations after were randomly presented separately, and none of the stimuli presented had any similar errors. Additionally, clarified methodology by adding a sentence that all stimuli presentations were specific-product neutral to avoid any product-specific inherent biases from participants

The three hypotheses are initially tested using 2 x paired t-test and one-way ANOVA. Given the within-subject, full-factorial design, a repeated measures multivariate ANOVA would seem suitable to test all three hypotheses and would also allow examination of potential interaction effects. Have the authors considered this? Interaction effects might also be added to the regression models. The potential interaction between total items and the number of cheaper items at the second store might be interesting to examine and could feed into Study 2.

RESPONSE: We did run an ANOVA analysis, but did not include total items, as we weren’t considering it a stimuli. A follow-up one-way ANOVA between the means of likelihood with ‘total items in cart’ equal to 19, 20, and 21 is insignificant. (p > .6). In a regression analysis, ‘total items in cart’ is insignificant, (p > .8) Both analysis show ‘total items in cart’ as insignificant, which is to be expected, as the total items was centered on 20, with the variation of +/- 1 to avoid order effect. 

Study 2

The decision to restrict analysis to a subset of the data would benefit from a fuller account and justification. The importance of why the total number of items is restricted between 5 and 22 could be clearer.

RESPONSE: Language to clarify the reason for the overlap-specific analysis to determine H4 significance has been added.

Study 3

No suggestions.

Discussion and Conclusion

The concluding section is informative and well-written, but could be strengthened in a number of ways.

First, there are multiple references to how “models” of consumer behavior need to take account of the quantity bias. It would be informative to refer explicitly to specific models here and include citations to relevant papers. Also, I am not unclear about the statement that such models should include a “distaste for single item shopping”. This could be construed to mean consumers prefer to buy multiple items in a single shopping trip rather than single items across multiple trips – something akin to weekly versus daily grocery shopping trips, which the paper does not examine.

RESPONSE: Our text does not make a good distinction between a specific model and research, in general. Upon rereading these paragraphs, we find that they are not necessary to the point of the paper and we have removed them. If the editors would like us to discuss research models that have looked at online shopping, but not included a consideration of how many items people are buying and how many different vendors they may need to buy those items from, then we would be happy to do that. However, it seems obvious in retrospect that our findings could be used by future researchers to develop more complete and better predictive models and we probably do not need to explicitly mention that in our paper.

Second, reference to experienced, remembered and anticipated utility is a little vague and would benefit from definition of these terms and citations to relevant papers. 

RESPONSE: Added clarification to utility and citation. 

Third, the possibility that people might discount time differently online vs physical world is an interesting. That individuals might be irrational in their valuation/use of their own time would benefit from linking to prior literature examining mental accounting in the context of time-denominated accounts, as opposed to money-denominated accounts. For example, when savings are in time rather than money, Leclerc et al. (1995) find evidence supporting the use of topical accounts, while findings in Frisch (1993) and Duxbury et al. (2005) suggest an absence of mental accounting effects for time

RESPONSE: Added to our discussion.

More generally, in all studies, the rationale offered for the regression analyses is to determine coefficient size. While these are reported in the tables, it would help the reader’s interpretation if the paper included brief discussion of the effect sizes for the behavioral biases across the three studies. 

RESPONSE: Effect size calculations included in methodology section and referenced in the discussion section.

Minor comments

1. On p5, “$28.0” in the main text should be “$28.01”. 

RESPONSE: Corrected.

2. Final sentence of first para on p.6 states researchers “determine” what consumers care about. But “determine” could be read as prescribe or dictate here, and I would recommend replacing by something more neutral such as “examine” or “evaluate”. 

RESPONSE: Replaced with ascertain.

3. Also on p.6, the reference to “single-item decision” in the context of the cited studies of mental accounting effects is not quite precise. Such studies use scenarios in which two items are to be purchased at a single store (in contrast to the possibility in the current paper that items will be purchased from multiple stores), as per the classic jacket and calculator problem on p.3, but where one is discounted at a second store. The non-discounted price of the original item is varied to manipulate the relative saving while holding absolute saving constant. This is necessary to isolate the separate effects of absolute and relative savings. Such a design holds constant the value of the absolute saving relative to the total combined expenditure, so there is not rational economic reason for differences in likelihood of travel. Hence, the phrasing in the paper needs to be more precise here

RESPONSE: We have removed the reference to a single item decision. 

4. On p.7, the closing speech marks on the quote from Thaler should not enclose the in-text citation to Thaler’s work. 

RESPONSE: Corrected.

5. There are occasional sentences where I would have liked to see more use of commas, but this is sometimes a matter of personal preference.

6. On p.22, first para, on first reading the phrase “greater total value of the shopping trip” might easily be misconstrued as referring to total cost of the basket, which renders the sentence somewhat meaningless. On closer consideration, I take the phrase to be referring to the value or utility derived from the savings to be made from traveling to the second store. Is this correct? Either way, the sentence needs rephrasing to avoid potential misinterpretation.

RESPONSE: Re-phrased.

7. In the reference for Mowen & Mowen (1986) the “*” can be removed from the title. 

RESPONSE: Corrected.

Reviewer #2

This paper defines a new behavioral bias in consumer behavior, named “quantity bias”, and

develops three studies to test its existence and measure its effect. Quantity bias means that

people value more the same savings (absolute and relative) when they spread among more

items.

I like the paper. It is relevant and original. The methodology is appropriate to test and measure

the effects. However there are some issues that should be addressed before publication.

• If I am not wrong, Study 1 (Figure 2) has two errors: the shopper would buy 5 items

(instead of 3) at store 2 (macaroni and cheese is less expensive at store 2) and the

savings would be 0.25 dollars (instead of 0.15 dollars) 

RESPONSE: Addressed as noted to reviewer #1.

• Figure 3. I don’t understand the numbers. To buy 19 items in total, I think the only

possibility is that the shopper buys 5x2=10 macaroni and pasta and 3x3=9 bread items

(there is not other way to get 19 as factor of 5 and 3). Then the shopper would spend

32.6 dollars in store 1. To save going to store 2 the shopper would buy the bread at

this store (bread is cheaper at store 2), but in this case the shopper would buy 3x3=9

items of bread at stores 2, not 1 item. May be I am wrong.

RESPONSE: Figure 3 is a representative specific-product free presentation that was utilized in the experiment. Presentations were distinct, and in random order, with language added to clarify. Additionally, specific product prices were not shown in those presentations, only potential savings and number of items (without specifying which items), and no indications that participants expected/perceived that presentations were tied to example setup scenario.

• How many presentations were shown to each subject? 4? I say this because in page 9

it is said 8 presentations. 

RESPONSE: It was eight repeated measures presentations, to give one presentation of each condition in the 2x2x2 design.

• Study 2. According to hedonic editing, the same saving is more valued when separated in more items. Then, H4 should be rejected, isn’t it?

RESPONSE: Our hypothesis is actually about having a fixed number of items at the second store and changing the number of items at the first store. In principle hedonic editing should not apply as the amount at store 2 is not changing. People should experience decreasing marginal utility for items at the first store, but the utility gained by going to the second store should remain constant. We believe you are correct about what hedonic editing would suggest. And you are correct in the sense that the hypothesis was rejected. 

To put the whole thing in terms of hedonic editing, the utility curve would have number of items on the X-axis and utility on the Y-Axis. Out hypothesis proposes that instead of number of items on the X-axis, people may have percent of items on the X-axis. Ultimately we are wrong in that, but that is what H4 is proposing. 

.

• P19: You say: “We conducted regression analysis of the likelihood of shopping at a

second store against the total number of items in the basket (relative effect)”

Shouldn’t the relative effect be instead # items bought at store 2 divided by # total

items bought?

RESPONSE: We have updated the language to be clearer as per our hypothesis that the relative effect was against the total number of items to be purchased at the second store.

• P19: You say: “Regression against the total number of items in the baskets was not

significant for any model, while # of items at the second store was significant. Thus, H3

(Quantity Bias) and H4 (Relative-Quantity) were not supported.” If # of items at the

second store was significant, doesn’t it imply that H3 is supported? 

RESPONSE: This was indeed a typo, as H3 was significant on the total sample, and has been corrected.

• Study 3 is for the online setting, and in my opinion is less relevant 

RESPONSE: Reviewer 1 found this to be an engaging follow-on experiment, and we feel others will find value in the offline vs online quantity bias effect being present.

For Both Reviewers

On page 2, the quote reference has been changed to properly reflect both Kahneman & Tversky 1984 and Thaler 1999, with reference citations updated.

On page 8, we have changed the example citation to an earlier work by Thaler, as the follow-on language is confusing from a timeline issue, and removed the references completely to the 2008 Thaler citation and from the references (page 33). Additionally, the word “two” was moved to its proper order in the quote.

On page 32, the Huang et al. reference was removed as it is not used in this version of the paper.

References were updated with the new citations for power analysis and the discussions on remembered/experienced utility and time-denominated mental accounts. 

For the comments from the editor.

The three requested files are named as requested.

Data is ready to be uploaded to a repository with notice of acceptance and will supply the DOI that is assigned.

Methods section updated with requested IRB consent.

Figure 1 has been updated with representative images to respect copyright issues, as no parts of the image have copyright associated, nor are any brand names referenced within the representative images.

References have been checked to ensure none have been retracted and updated as noted to the reviewers.

Awaiting final acceptance to submit reformatted article to PLOS standards (including the uploading of figures as indicated in your submission requirements section).

---

## [Editor Report · Decision Letter 1]

19 Jan 2022

Quantity Bias in comparison-shopping of multi-item baskets

PONE-D-21-23401R1

Dear Dr. Niswanger,

We’re pleased to inform you that your manuscript has been judged scientifically suitable for publication and will be formally accepted for publication once it meets all outstanding technical requirements.

Kind regards,

James H. Cardon, Ph.D.

Academic Editor

PLOS ONE

Additional Editor Comments (optional):

Dear Ross,

I have read your revised manuscript and your responses to comments of both reviewers. Thank you for your attention to their comments and suggestions. The paper is much improved. Thank you for submitting to PlosOne.

James Cardon

Department of Economics

Brigham Young University
---

## [Editor Report · Acceptance letter]

26 Jan 2022

PONE-D-21-23401R1 

Quantity Bias in comparison-shopping of multi-item baskets 

Dear Dr. Niswanger:

I'm pleased to inform you that your manuscript has been deemed suitable for publication in PLOS ONE. Congratulations! Your manuscript is now with our production department. 

Kind regards, 

on behalf of

Dr. James H. Cardon 

Academic Editor

PLOS ONE